# Effects of Drinking Water Temperature and Flow Rate during Cold Season on Growth Performance, Nutrient Digestibility and Cecum Microflora of Weaned Piglets

**DOI:** 10.3390/ani10061048

**Published:** 2020-06-18

**Authors:** Zhenyu Zhang, Zeqiang Li, Hua Zhao, Xiaoling Chen, Gang Tian, Guangmang Liu, Jingyi Cai, Gang Jia

**Affiliations:** 1Animal Nutrition Institute, Sichuan Agricultural University, Chengdu 611130, Sichuan, China; zhangzheny1111@126.com (Z.Z.); lizeqiang896527762@126.com (Z.L.); zhua666@sicau.edu.cn (H.Z.); xlchen@sicau.edu.cn (X.C.); 13555@sicau.edu.cn (G.T.); liugm@sicau.edu.cn (G.L.); 11890@sicau.edu.cn (J.C.); 2Institute of Animal Husbandry and Veterinary Medicine, Meishan Vocational Technical College, Meishan 620010, Sichuan, China

**Keywords:** water parameters, winter, pigs, growth performance, nutrient utilization, microflora

## Abstract

**Simple Summary:**

Water is an essential nutrient pigs need to sustain life and ensure growth. Determining the appropriate drinking water supply parameters during cold weather are critical for the welfare and growth of pigs, especially vulnerable weaned piglets. This study explored different combinations of drinking water temperature (DWT) and flow rate (DWF) for weaned piglets during winter. It measured their growth performance, nutrient digestibility and cecum microbial diversity, aiming to figure out the optimal water supplying conditions for weaned piglets. The results indicated that a combination of DWT of 30 °C and DWF of 300 mL/min decreased diarrhea occurrence. Furthermore, this increased growth performance and nutrient digestibility, accompanied by improvement of the dominant cecum microflora, mainly manifested in a reduced abundance of Proteobacteria and increased abundance of Bacteroidetes. This study enriches our understanding of the connection between water supply, growth performance and cecum microbiota on weaned piglets during the cold season.

**Abstract:**

Although water is one of the most important nutrients and is essential for various physiological processes within the body, it does not receive adequate consideration when ensuring optimal nutrition and growth performance in piglets. This study was conducted to investigate the effects of drinking water temperature (DWT) and flow rate (DWF) on growth performance, nutrient digestibility and cecum microflora in weaned piglets during cold weather. Sixty-four piglets with an average body weight of 8.60 ± 0.5 kg were allotted into four groups with four replicates in each group and four pigs in each replicate. The DWT and DWF were set for each group as follows: (1) 13 °C + 300 mL/min, (2) 13 °C + 700 mL/min, (3) 30 °C + 300 mL/min and (4) 30 °C + 700 mL/min, respectively. All groups were fed the same diet during the 28 d trial. The body weight at day 0 and day 29, as well as daily feed intake, were recorded. Diarrhea severity was assessed every day. Fresh fecal samples were collected for four consecutive days at the end of the experiment for the digestibility test. Cecum content was collected after sacrifice for microbial composition analysis. The results indicated that: (1) DWT at 30 °C promoted the average daily gain (ADG) of weaned piglets considerably (*p* = 0.043) and decreased feed to weight ratio when compared with DWT at 13 °C (*p* = 0.045). DWF had no substantial effect on the growth performance of piglets (*p* > 0.05). (2) The 30 °C DWT groups had higher apparent digestibility of crude protein, crude fat and energy than the 13 °C DWT groups (*p* < 0.05), while DWF had no significant effect on the apparent digestibility of nutrients (*p* > 0.05). (3) DWT at 30 °C increased the Bacteroidetes abundance and decreased the Proteobacteria abundance in cecum digesta. The change in these two factors may be related to a decrease in diarrhea and the improvement of growth performance. Different DWF had no substantial effect on the cecum microbial structure. To sum up, providing a DWT of 30 °C to weaned piglets in cold weather reduced the abundance of harmful bacteria in the cecum and improved the apparent nutrient digestibility, which is beneficial for maintaining a healthy intestinal microenvironment and promoting growth performance. A lower DWF of 300 mL/min had no adverse effect on growth performance. Therefore, a combination of 30 °C + 300 mL/min is recommended for weaned piglets during cold weather for the consideration of animal welfare and production efficiency.

## 1. Introduction

Water is necessary for several physiological activities and plays a crucial role in regulating body temperature, promoting digestive tract development and maintaining the health of the urinary system [1]. Although water is the nutrient most consumed by animals, little research has been carried out to explore the appropriate supply parameters, such as drinking water temperature (DWT) and flow rate (DWF), for weaned animals. Inappropriate DWT and DWF may have an adverse effect on them. After weaning, the water source of young animals is switched from warm, slowly flowing breast milk to room temperature, fast-flowing tap water, which can cause stress during the cold season. Cold water stress, together with feed transition, environmental stress and social stress, can affect the gastrointestinal tract health of young animals, leading to poor digestibility and growth performance [2].

Gastrointestinal tract health is known to be largely affected by the composition of intestinal microflora [3]. The intestinal microflora is a complex system and it is increasingly being accepted as a factor that shapes host metabolism and health [4]. Recent studies have found that the intestinal microflora in pigs play a crucial role in nutrient processing and energy harvesting [5,6], which, in turn, leads to variation in feed efficiency.

In these contexts, improving current knowledge of how DWT and DWF can impact the intestinal digestion and microflora profile could be of high relevance for the welfare of weaned animals. Thus, this study provided different combinations of DWT and DWF to weaned piglets during the cold season, and measured the growth performance, nutrient digestibility and cecum microbial diversity, aiming to figure out the appropriate DWT and DWF for weaned piglets and enhance the understanding of their association with intestinal microflora.

## 2. Materials and Methods

All of the experimental procedures that were used in this research were in accordance with the ARRIVE guidelines and the Regulation on the Administration of Laboratory Animals (2017, China State Council). The Institutional Animal Care and Use Committee of Sichuan Agricultural University approved these procedures (SCAUAC201711-2).

### 2.1. Animals and Experimental Design

Sixty-four 30-d-old DLY (Duroc × Landrace × Yorkshire) weaned piglets (8.6 ± 0.5 kg, mean ± SD) were randomly divided into four groups with four replicates in each group and four pigs in each replicate, in a balanced sex ratio (2 males and 2 females). Each replicate was kept in an individual pen (2.5 m × 3 m) under temperature-controlled conditions (22–25 °C) with a 12 h: 12 h light–dark cycle for 28 days. Different DWT and DWF were provided for each group: (1) 13 °C + 300 mL/min, (2) 13 °C + 700 mL/min, (3) 30 °C + 300 mL/min and (4) 30 °C + 700 mL/min, respectively. All groups were fed the same diet for starting pigs according to the nutritional requirements recommended by NRC [7]. Each pen contained two pig waterers (6.8 cm × 2 cm. stainless steel, Model WB20XH, Zhenhaidongyi Machinery co. LTD, Ningbo, China). The piglets were provided with free access to feed and drinking water.

### 2.2. Feeding and Diarrhea Occurrence

At the start and end of the experiment, pigs were weighed in each pen as a unit. Feed consumption was recorded daily. The average daily feed intake (ADFI), average daily gain (ADG) and feed-to-gain ratio (F:G) were calculated. The feces of all pigs were scored every day, and the diarrhea severity was accessed according to Marquardt et al. [8]. The pigs with diarrhea were identified to avoid double counts. Diarrhea rate was calculated by dividing the number of pigs with diarrhea by the total number of pigs in each treatment. The diarrhea index was calculated as (∑fecal scores for the duration of study)/*n* [9].

### 2.3. Sample Collection

Samples of the pelleted feed were taken after its production and stored at −20 °C for later digestibility determination.

At day 25–28, fresh fecal samples were collected from 4 different locations in each pen floor and mixed in a sterile plastic bag. Ten milliliters of 10% hydrochloric acid was added to 100 g of feces to prevent the ammonia nitrogen evaporation. The mixture was then kept at −20 °C for the digestibility test.

On day 29, one pig closest to the average pen weight was chosen from each replicate (sixteen pigs in total). The pigs were anesthetized by a combination of azaperone (2 mg/kg body weight) and ketamine (25 mg/kg body weight) administered intramuscularly. Then the abdominal cavity was quickly opened. After all the intestinal segments were separated, 0.5 g cecum content of each piglet was collected and put in sterile EP tubes. The samples were stored at −80 °C for DNA extraction. 

### 2.4. Apparent Digestibility of Nutrients

Acid-insoluble ash (AIA) was used as an inert marker to determine the apparent digestibility [10]. Samples were air-dried at 65 °C to a constant weight and crushed through a 40-mesh sieve. Then they were analyzed for crude protein (CP), crude ash and ether extract (EE) according to the following procedures from the Association of Official Analytical Chemists [11]: CP (Method 990.03), ash (Method 942.05) and EE (Method 920.39). The gross energy (GE) was measured in an automatic oxygen bomb calorimeter (PARR 6400; PARR Instruments CO., Moline, IL, US). The apparent nutrient digestibility was calculated according to Bovera et al. [12].

### 2.5. Microflora Analysis

A total of 0.3 g cecum digesta was thawed and employed to extract genomic DNA using the QIAamp DNA Stool Mini Kit (Qiagen, Valencia, USA) following the manufacturer’s instructions. The integrity and quantification of DNA were determined by agarose gel electrophoresis and a NanoDrop ND-2000 spectrophotometer (Thermo Fisher, USA), respectively. The universal primer sequence, 515F (5′-GTGYCAGCMGCCGCGGTAA-3′) and 806R (5′-GGACTACHVGGGTWTCTAAT-3′), were used to amplify the V4 region of the 16S rDNA gene. PCR was performed with the following amplification conditions: 95 °C for 2 min; 25 cycles at 95 °C for 1 min, 55 °C for 1 min, 72 °C for 1 min and 72 °C for 5 min. The sequence readings were processed on the QIIME v1.9.1 pipe [13] using the default settings. Readings were clustered into operational taxonomic units (OTUs) with 97% sequence similarity and picked by the subsampling open reference approach at 10% of sequences subsampled. Representative sequences were assigned to a classification of bacteria 16S GreenGenes v.13.8 reference database [14] according to the 90% confidence threshold, and sequence alignment was obtained through uclust. Chimera Slayer was employed to remove the chimeric sequence, and singletons and OTUs with a relative abundance of less than 0.005% in all samples were removed, according to Bokulich et al. [15]. The Vegan R package was used to calculate the alpha-diversity indexes, including Chao I, Simpson, Shannon and phylogenetic diversity (PD) whole tree. Principal component analysis was used to reduce the dimensions of the cecum microflora data.

### 2.6. Statistical Analysis

The statistical calculations were analyzed using the SPSS software package (SPSS vs. 24, SPSS Inc., Chicago, IL, USA). The statistical difference of normally distributed data, including growth performance, diarrhea rate and nutrient digestibility, were analyzed by using two-way ANOVA. The relative abundance of microbial phyla, genera and OTUs were determined using the nonparametric Kruskal–Wallis test. All *p*–values of the bacterial community obtained by the nonparametric Kruskal–Wallis test were corrected for the false discovery rate. A value of *p* < 0.05 was considered significant.

These sequence data were submitted to the NCBI database under accession number PRJNA630063.

## 3. Results

As no significant interaction was detected in the parameters of Table 1, Table 2, Table 3 and Table 4, results will be described by the main effects.

### 3.1. Growth Performance and Diarrhea Occurrence

Data on the effects of DWT and DWF on growth performance are presented in Table 1. DWT at 30 °C showed a trend of increasing the final body weight (BW; *p* = 0.065) and promoted the ADG of weaned piglets considerably (*p* = 0.043) compared to the 13 °C groups. Additionally, it reduced the F: G ratio (*p* = 0.045), but no substantial effect was detected on ADFI (*p* = 0.636). 

Low DWT caused a high incidence of diarrhea (Table 2). The diarrhea rate and index of weaned piglets in the 13 °C groups were significantly higher than those in the 30 °C groups (*p* = 0.001 and *p* = 0.000, respectively). Compared with DWT at 13 °C, DWT at 30 °C reduced the diarrhea rate of weaned piglets approximately by 50.00% and the diarrhea index by 38.14%. There was no substantial effect of DWF on the diarrhea rate (*p* = 0.380).

### 3.2. Nutrient Digestibility

The DWT at 30 °C promoted the apparent digestibility of CP (*p* = 0.000), EE (*p* = 0.035) and energy (*p* = 0.007), but had no substantial effect on the digestibility of crude ash (*p* = 0.266; Table 3). The DWF had no significant effect on the digestibility of CP, EE, crude ash or energy (*p* > 0.05). 

### 3.3. Sequencing, Richness and Diversity of the Bacterial Community

The dilution curve gradually flattened with the increase of sequencing depth, indicating reasonable sequencing quantities (Figure 1). A total of 1377, 1762, 1052 and 1263 OTUs were obtained in the 13 °C + 300 mL/min, 13 °C + 700 mL/min, 30 °C + 300 mL/min and 30 °C + 700 mL/min group, respectively (Figure 2).

The indexes of Chao 1, Simpson, Shannon and Faith’s PD, which all indicate species richness, were calculated in order to evaluate the alpha diversity in different ways (Table 4). The indexes of Chao1, Faith’s PD and Shannon were not affected by DWT (*p* > 0.05), and were increased with the rise of DWF (*p* < 0.05). The Simpson index was not affected by either DWT or DWF (*p* > 0.05). There was no substantial effect of DWT and DWF interaction on the alpha diversity index (*p* > 0.05).

The principal coordinate analysis showed that samples of the same treatment could be well clustered, indicating the results were meaningful (Figure 3). Different processing groups were scattered in far apart locations on the coordinate system. The distance between the 13 °C + 300 mL/min group and the 13 °C + 700 mL/min group was relatively close. Meanwhile, the distance between the 30 °C + 300 mL/min group and the 30 °C + 700 mL/min group was near, indicating there is a relatively high similarity of microbial structure within the same DWT treatments. In comparison, when the DWF was at 700 mL/min, the aggregation degree of samples under the same DWF treatments was low, indicating that the microbial structure within the group was significantly different.

### 3.4. Microbiome Modulation at Taxonomic Levels

Figure 4A shows the top twelve bacterial phylum in the cecum digesta. Firmicutes constituted the most prevalent phylotype, comprising over 57.0% of the cecum microbial population, followed by Proteobacteria, Actinobacteria, Acidobacteria, Chloroflexi, Thaumarchaeota, Spirochaetae, Deferribacteres, Nitrospirae, Euryarchaeota and Verrucomicrobia. DWT at 30 °C increased the relative abundance of Bacteroidetes, and decreased the relative abundance of Proteobacteria, Actinobacteria, Acidobacteria, Chloroflexi and Thaumarchaeota (*p* < 0.05). DWF had no substantial effect on the relative abundance of microbes. There was no significant interaction between DWT and DWF on microbial phylotype (*p* > 0.05).

At the genus level, the prevalent phylotypes were *Megasphaera, Lactobacillus, Anaerovibrio, Alloprevotella, Prevotella* 9, *Pseudobutyrivibrio, Phascolarctobacterium, Faecalibacterium, Acinetobacter, Sphingomonas, Prevotellaceae* NK3B31 group and *Selenomonas* (Figure 4B,C). The DWT change had no substantial effect on the relative abundance of *Megasphaera, Pseudobutyrivibrio, Phascolarctobacterium* and *Acinetobacter* (*p* > 0.05). The DWT at 30 °C decreased the relative abundance of *Lactobacillus* and *Sphingomonas*, and improved the relative abundance of *Anaerovibrio, Alloprevotella, Prevotella* 9, *Faecalibacterium* and *Prevotellaceae* NK3B31(*p* < 0.05). DWF had no significant effect on the relative abundance of microbes at the genus level (*p* > 0.05).

## 4. Discussion

After weaning, the water source of piglets changed from warm breast milk to cold tap water, which can easily cause diarrhea and reduce growth performance. When DWT is lower than the body temperature of pigs, extra energy is needed to warm up the incoming water. A previous study found that pigs provided with a DWT of 30 °C had a significantly higher growth performance compared with the ones provided with a DWT of 5 °C [16]. Similar results were obtained in the present study, with ADG increased by 10.00% and F:G decreased by 8.90% after DWT treatment at 30 °C. Therefore, the piglets do not need to expend extra energy to heat up the water, and more energy can be used to increase their growth performance.

In addition to ambient temperature management, DWF was also important for the growth performance of piglets [17]. Nienber et al. [16] found that the effect of DWF was in close relation with the environmental temperature. At an ambient temperature of 35 °C, when the DWF was 100 mL/min and 1100 mL/min, the average ADG of pigs was 0.278 kg/d and 0.466 kg/d, respectively. While at an ambient temperature of 5 °C, the average ADG of pigs decreased from 0.855 to 0.730 kg/d under the same above mentioned DWF increase. These results suggest that the effect of DWT and DWF on livestock should be considered together. In the present study, no significant interaction was observed between DWT and DWF on the weaned piglet’s growth performance, which may be due to different DWT and DWF conditions in these two studies.

Sudden changes in the environment and feed after weaning can stress piglets and increase diarrhea occurrence. The 13 °C groups had a higher diarrhea index than the 30 °C groups, which may be due to the increased sympathetic nerve excitability and exacerbated bowel motility caused by drinking cold water [18,19].

The total bacterial community that inhabits the gastrointestinal tract of mammals numbers at about 10^14^, which consists of 500–1000 kinds of bacteria [20]. Under the combined action of these bacteria, the intestinal tract maintains a relatively stable microenvironment. Firmicutes and Bacteroidetes are the primary microflora in the intestine, accounting for more than 98% of all the bacteria, while the proportion of other phylum is only about 1% [21,22]. Healthy microflora plays a vital role in intestinal development, nutrient digestion absorption and immunity. Weaned piglets are susceptible to the influence of the external environment due to their immature physiological functions, resulting in the imbalance of intestinal flora and diarrhea. Bacteroidetes participate in carbohydrate fermentation, metabolism of polysaccharide, bile acid and steroid, and play an essential role in maintaining normal intestinal physiology and body health [23]. Grover et al. [24] found that the content of Bacteroidetes in the intestinal tracts of people with diarrhea was significantly lower than in the intestinal tracts of healthy ones. The present study showed that DWT at 13 °C reduced the relative abundance of Bacteroidetes, which may be one of the causes of diarrhea. Proteobacteria is a large group of bacteria, including many pathogenic members, such as *Escherichia coli*, *Salmonella*, *Vibrio cholerae* and *Helicobacter pylori* [25]. The relative abundance of Proteobacteria is known to be negatively related to the health status of its host. A small amount of Proteobacteria can coexist peacefully with the host, but as the number of Proteobacteria increases, it can cause a series of diseases. Shin et al. [26] found that the abundance of Proteobacteria in the intestines of healthy people was lower, while it increased significantly in diarrhea patients, and was accompanied by the increase of proinflammatory factor secretion. Durban et al. [27] also found that Proteobacteria may be involved in the occurrence of acute diarrhea. This is consistent with the results of the present study. DWT at 30 °C decreased the relative abundance of Proteobacteria in the cecum digesta of weaned piglets, accompanied by a reduced diarrhea rate and diarrhea index. Actinobacteria also plays an important role in maintaining animal health [28]. It has abilities to provide essential vitamins and amino acids, protecting the intestinal barrier, acidifying the intestinal microenvironment, inhibiting the growth of pathogenic bacteria and reducing the production of endotoxin. The present study suggested a decrease in the abundance of Actinobacteria in piglets provided with DWT at 30 °C. It may be due to the complexity of the intestinal micro-ecosystem. Although the number of Actinobacteria went down, the abundance of other beneficial bacteria increased. Under their joint actions, the intestinal micro ecological balance was maintained and exhibited beneficial effects on growth performance.

Recent studies have found that Acidobacteria may be associated with the degradation of cellulose [29,30], and Chloroflexi may be associated with the degradation of carbohydrates and amino acids [31]. In the present study, DWT at 13 °C increased the abundance of Acidobacteria and Chloroflexi in the cecum digesta. One of the possible reasons is that feed is not fully digested in the small intestine, and the cecum digesta are still rich in nutrients, which is conducive to Chloroflexi survival. 

*Anaerovibrio* is involved in lipid degradation, and its relative abundance grows as the dietary fat ratio increases [32]. *Alloprevotella* can produce short-chain volatile fatty acids, and its abundance in patients with metabolic syndrome and diabetes is significantly reduced [33,34]. In this study, DWT at 30 °C improved the abundance of *Anaerovibrio* and *Alloprevotella*, which may increase the digestibility of crude fat.

Although piglets provided DWT at 30 °C had a low relative abundance of *Lactobacilli* than that of provided DWT at 13 °C, DWT at 30 °C increased the relative abundance of Firmicutes and Bacteroidetes, and decreased the Proteobacteria abundance, compensating for the decrease in *Lactobacilli*. Under the joint action of these various microbes, a balanced intestinal micro-ecology was maintained.

## 5. Conclusions

Overall, the present data showed that the DWT should be set at 30 °C and DWF at 300 mL/min for weaned piglets in cold environments. This combination of DWT and DWF is beneficial for improving growth performance and feed efficiency, as well as decreasing diarrhea index in weaned pigs. These beneficial effects may be due to changes in microbial community compositions of the cecum, that led the weaned piglets to have higher utilization efficiency of energy, protein and fat in the feed. Therefore, the combination of drinking water parameters proposed in this study can be used to improve the body weight and health status of weaned piglets in future pig farm management during the cold season.

## Figures and Tables

**Figure 1 animals-10-01048-f001:**
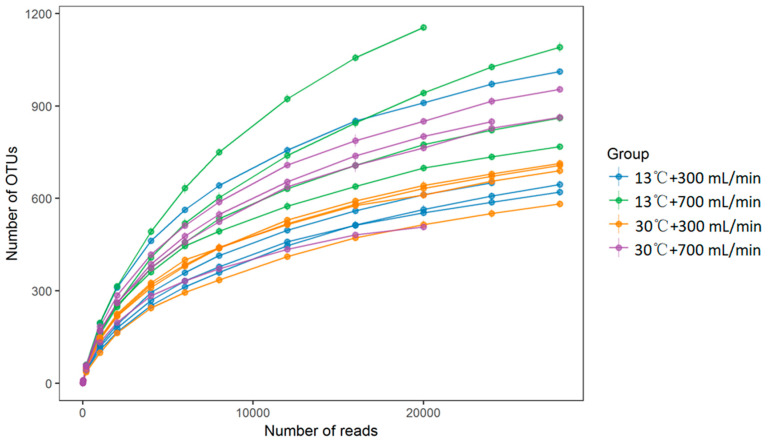
Rarefaction curves based on operational taxonomic units (OTUs) in the cecum digesta.

**Figure 2 animals-10-01048-f002:**
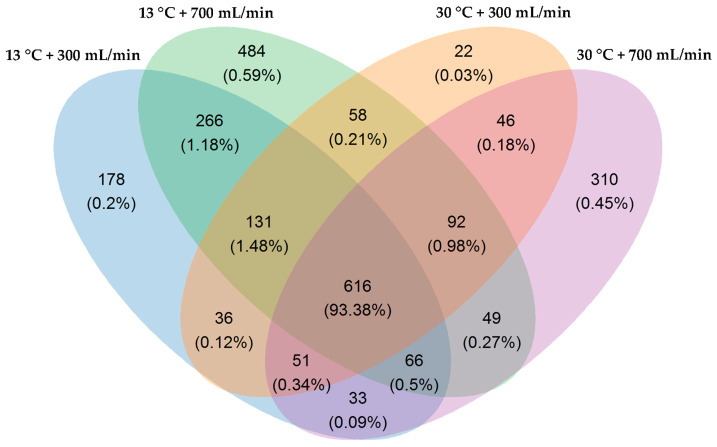
Venn diagrams of shared genes.

**Figure 3 animals-10-01048-f003:**
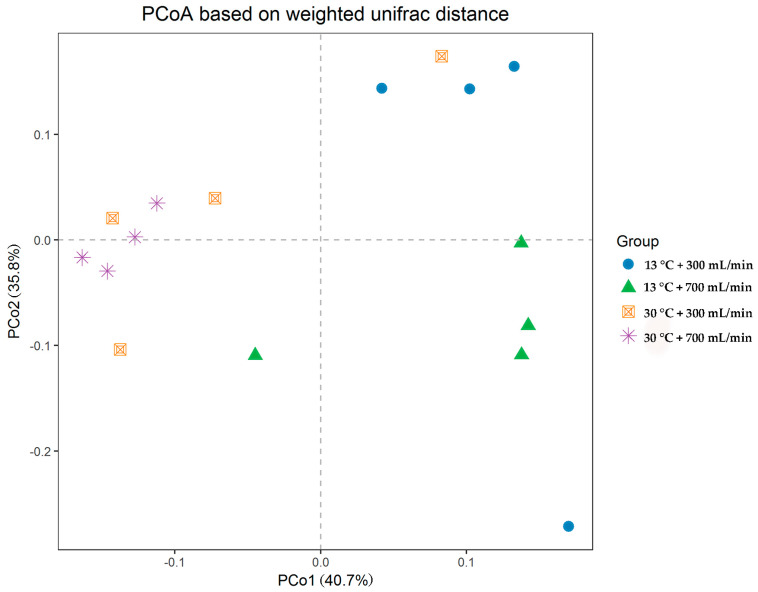
Principal co-ordinates analysis based on the weighted Unifrac metric of cecal microflora among all samples.

**Figure 4 animals-10-01048-f004:**
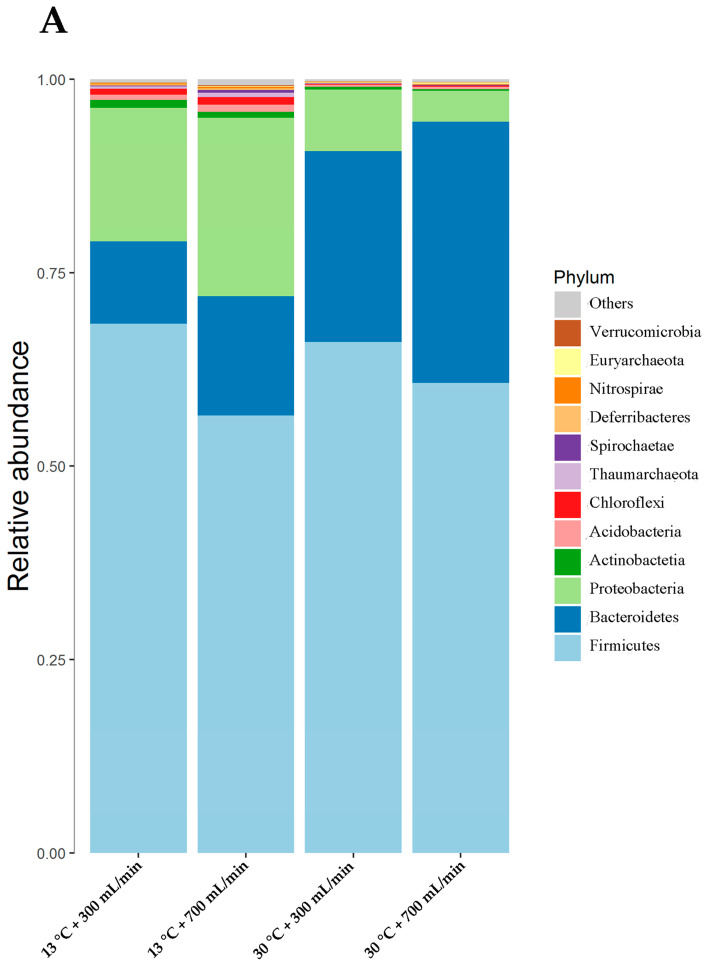
The relative abundance of the top twelve phyla (**A**) and genera (**B**) in the cecum digesta. (**C**) Boxplot at the genus level. * represents *p* < 0.05.

**Table 1 animals-10-01048-t001:** Effects of drinking water temperature and flow rate on the growth performance of weaned piglets ^1^.

Items	DWT	13 °C		30 °C		*p*-Value		
	DWF	300 mL/min	700 mL/min	300 mL/min	700 mL/min	DWT	DWF	DWT × DWF
Initial BW (kg)		8.60 ± 0.14	8.60 ± 0.24	8.60 ± 0.18	8.60 ± 0.08	1.000	1.000	1.000
Final BW (kg)		20.54 ± 1.07	18.95 ± 1.57	20.79 ± 0.64	21.04 ± 1.12	0.065	0.267	0.139
ADG (kg/d)		0.43 ± 0.04	0.37 ± 0.05	0.44 ± 0.02	0.44 ± 0.04	0.043	0.220	0.103
ADFI (kg/d)		0.65 ± 0.06	0.63 ± 0.03	0.65 ± 0.04	0.66 ± 0.06	0.636	0.790	0.583
F: G		1.53 ± 0.05	1.73 ± 0.25	1.49 ± 0.04	1.48 ± 0.02	0.045	0.168	0.124

^1^ Values are means ± SEM (*n* = 4); DWT = drinking water temperature; DWF = drinking water flow rate; BW = body weight; ADG = average daily gain; F: G = the ratio of feed to gain.

**Table 2 animals-10-01048-t002:** Effects of drinking water temperature and flow rate on the diarrhea rate and index of weaned piglets ^1^.

Items	13 °C		30 °C		*p*-Value		
	300 mL/min	700 mL/min	300 mL/min	700 mL/min	DWT	DWF	DWT × DWF
Diarrhea rate (%)	16.07 ± 2.31	17.63 ± 1.98	7.59 ± 4.28	9.37 ± 5.12	0.001	0.380	0.953
Diarrhea index	0.55 ± 0.04	0.63 ± 0.06	0.32 ± 0.06	0.41 ± 0.08	0.000	0.024	0.917

^1^ Values are means ± SEM (*n* = 4); DWT = drinking water temperature; DWF = drinking water flow rate.

**Table 3 animals-10-01048-t003:** Effects of drinking water temperature and flow rate on the apparent nutrient digestibility ^1^.

Items	13 °C		30 °C		*p*-Value		
	300 mL/min	700 mL/min	300 mL/min	700 mL/min	DWT	DWF	DWT × DWF
CP (%)	65.78 ± 2.38	67.21 ± 3.02	71.81 ± 2.21	72.96 ± 1.30	0.000	0.287	0.906
Ash (%)	48.05 ± 3.22	46.39 ± 1.67	48.85 ± 3.17	49.56 ± 1.81	0.266	0.785	0.499
EE (%)	56.60 ± 4.58	52.13 ± 3.22	59.80 ± 1.80	58.42 ± 5.44	0.035	0.169	0.456
Energy (%)	74.89 ± 2.30	73.54 ± 2.30	77.44 ± 0.98	76.99 ± 1.51	0.007	0.354	0.640

^1^ Values are means ± SEM (*n* = 4); DWT = drinking water temperature; DWF = drinking water flow rate; CP = crude protein; EE = ether extract.

**Table 4 animals-10-01048-t004:** Effects of drinking water temperature and flow rate on the alpha diversity of cecum microflora ^1^.

Items	13 °C		30 °C		*p*-Value		
	300 mL/min	700 mL/min	300 mL/min	700 mL/min	DWT	DWF	DWT × DWF
Chao ^1^	885.12 ± 185.78	1212.21 ± 239.51	818.24 ± 946.79	946.79 ± 218.81	0.110	0.036	0.323
Shannon	4.13 ± 0.50	4.71 ± 0.22	4.19 ± 0.58	4.62 ± 0.32	0.960	0.035	0.715
Simpson	0.93 ± 0.02	0.97 ± 0.01	0.93 ± 0.06	0.96 ± 0.01	0.906	0.072	0.963
PD	48.76 ± 11.34	65.22 ± 13.85	43.47 ± 3.69	50.92 ± 8.32	0.075	0.035	0.387

^1^ Values are means ± SEM (*n* = 4); DWT = drinking water temperature; DWF = drinking water flow rate.

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
