# Peer review of "Effects of Drinking Water Temperature and Flow Rate during Cold Season on Growth Performance, Nutrient Digestibility and Cecum Microflora of Weaned Piglets"

_animals, 2020, doi:10.3390/ani10061048_

Round 1

Reviewer 1 Report

The trial had the objective to evaluate drink water temperature and flow rate for weaned piglets on the growth performance, nutrient digestibility, and cecum microbial diversion.

Line 2: you should modify the title and add the information about the cold weather.

Line 17: “…water feeding conditions for weaned…” change feeding for supplying.

Line 49: economic benefits? You can´t say it! You did not conduct an economic evaluation. Neither evaluate waste water…

Line 50: Do not repeat words used on the title. All the words are the same you must change, as it will enable the paper to be found easier.

Line 59: “…warm, bubbling breast…” Bubbling what you mean with this?

Line 140-141: “…Interestingly, a higher DWF at 700 mL/min with a set temperature of 13 °C decreased the final BW, ADG and ADFI (p < 0.05).” à This information is not correct! You did not observe an interaction! If, you check table 1 you had effects just for DWT, so you should group the averages of 300 and 700 mL and discuss just the temperature!

Table 1: You should group the results and the letter should be included just in the average of the groups not in the four average as you did not have an interaction à you must take this in account for all the tables. Final BW was 0.065 this is not different! You can´t present letters!

Line 146-148: “…The diarrhea rate and diarrhea index of weaned piglets in the 13 °C + 300 mL/min group and 13 °C + 700 mL/min group were significantly higher than those in the 30 °C + 300 mL/min group and 30 °C + 700 mL/min group (p < 0.05).” à you should group the results and discuss the average of ([0.16+0.18])/2 for 13ºC group) and ([0.08+0.09])/2 for 30ºC group), you di not have an interaction, so your comparison is just for DWT for Diarrhea rate!

Line 149: What is DMT? It is DWT?

Line 149: This is the only comparison that you can do, as the only difference was for DWT!

Table 2: Diarrhea rate (%) à it is not presented in percentage, you must multiply by 100 to be in percent.

Line 158-161: This results presentation are correct.

Table 3: you should group the results and the letter should be included just in the average of the groups not in the four average as you did not have an interaction à you must take this in account for all the tables.

Table 4: you should group the results and the letter should be included just in the average of the groups not in the four average as you did not have an interaction à you must take this in account for all the tables.

You must improve the discussion section.

Line 230-232: “…In the present study, the DWT at 13 °C with a higher DWF led to lower growth performance, whereas the DWT at 30 °C with a higher DWF promoted the growth performance …” This information is not correct you did not have an interaction.

Line 236-237: “…It is worth noting that the 13 °C + 700 mL/min group had a higher diarrhea index than the 13 °C + 300 mL/min group, which may be caused by the greater cold stimulation of the gastrointestinal tract caused by a higher DWF. …” This information is not correct you did not have an interaction.

Conclusion: You should present the conclusion, not repeat the results, just say: Based in our results, piglets weaned in cold environments (winter) the water temperature should be set in 30 °C and water flow in 300 mL/min.

Line 291: You did not evaluate water waste, you cannot conclude it!

Author Response

Thank you very much for the comments and suggestions. We have modified the
manuscript accordingly.

Response to Reviewer 1 Comments

Title: you should modify the title and add the information about the cold weather.

Response: Thank you for pointing out that the title missed the keywords. The information of "during the cold season" has been added in the Title (line 3).

Line 17: “…water feeding conditions for weaned…” change feeding for supplying.

Response: “optimal water feeding” was revised as “optimal water supplying” in line 17.

Line 49 economic benefits? You can´t say it! You did not conduct an economic evaluation. Neither evaluate waste water…

Response: “for the consideration of animal welfare, economic benefits and water waste reduction” has been revised as “for the consideration of animal welfare and production efficiency” in line 49.

Line 50: Do not repeat words used on the title. All the words are the same you must change, as it will enable the paper to be found easier.

Response: Thanks so much for this suggestion. We have substituted the keywords “Water parameters; Winter; Pigs; Nutrient utilization” for the previous keywords “Drinking water temperature; Drinking water flow rate; Weaned piglets; Nutrients digestibility”.

Lines 58-59: “…warm, bubbling breast…” Bubbling what you mean with this?

Response: Sorry for the misuse of word. “warm, bubbling breast milk ” has been substituted with “warm, slowly flowing breast milk”.

Line 150: “…Interestingly, a higher DWF at 700 mL/min with a set temperature of 13°C decreased the final BW, ADG and ADFI (p < 0.05).” à This information is not correct! You did not observe an interaction! If, you check table 1 you had effects just for DWT, so you should group the averages of 300 and 700 mL and discuss just the temperature!

Response:  “…Interestingly, a higher DWF at 700 mL/min with a set temperature of 13°C decreased the final BW, ADG and ADFI (p < 0.05).” has been removed.

Table 1: You should group the results and the letter should be included just in the average of the groups not in the four average as you did not have an interaction à you must take this in account for all the tables. Final BW was 0.065 this is not different! You can´t present letters!

Response: Thanks for the guidance. We realized the main effect of DWF was not significant and there was no substantial interaction between DWT and DWF, therefore the superscript letters have been deleted in Table 1. 

Line 155: “…The diarrhea rate and diarrhea index of weaned piglets in the 13°C + 300 mL/min group and 13°C + 700 mL/min group were significantly higher than those in the 30°C + 300 mL/min group and 30°C + 700 mL/min group (p < 0.05).” à you should group the results and discuss the average of ([0.16+0.18])/2 for 13ºC group) and ([0.08+0.09])/2 for 30ºC group), you did not have an interaction, so your comparison is just for DWT for Diarrhea rate!

Response: The original sentence has been revised as “The diarrhea rate and index of weaned piglets in the 13°C groups were significantly higher than those in the 30°C groups ” in lines 155-156

Line 157: What is DMT? It is DWT?

Response: Sorry for the spelling error. The misspelled abbreviation DMT has been changed to DWT throughout the manuscript.

Line 157: This is the only comparison that you can do, as the only difference was for DWT!

Response: The description of Table 2 now mainly focus on the influence of DWT on diarrhea occurrence, and the original description of DWF: “but DWF at 700 mL/min increased the diarrhea index (p < 0.05) when compared with the 300 mL/min groups. There was no substantial interaction between DWT and DWF on the diarrhea index (p > 0.05)”, has been changed to “There was no substantial effect of DWF on the diarrhea rate (p = 0.380)”, in lines 158-159.

Table 2: Diarrhea rate (%) à it is not presented in percentage, you must multiply by 100 to be in percent.

Response: Diarrhea rate has been multiplied by 100 in Table 2 to give the right presentation.

Line 163-165: This results presentation are correct.

Response: Thanks for the confirmation.

Table 3: you should group the results and the letter should be included just in the average of the groups not in the four average as you did not have an interaction à you must take this in account for all the tables.

Response: The superscript letters have been deleted in Table 3 and the corresponding result description has been changed to “The DWT at 30°C promoted the apparent digestibility of CP (p = 0.000), EE (p = 0.035) and energy (p = 0.007), but had no substantial effect on the digestibility of crude ash (p = 0.266) (Table 3). The DWF had no significant effect on the digestibility of CP, EE, crude ash or energy (p > 0.05)” in lines 163-165.

Table 4: you should group the results and the letter should be included just in the average of the groups not in the four average as you did not have an interaction à you must take this in account for all the tables.

Response: The superscript letters have been deleted in Table 4

You must improve the discussion section.

Line 232-234: “…In the present study, the DWT at 13°C with a higher DWF led to lower growth performance, whereas the DWT at 30°C with a higher DWF promoted the growth performance …” This information is not correct you did not have an interaction.

Response: This sentence has been changed to “…In the present study, no significant interaction was observed between DWT and DWF on the weaned piglet's growth performance, which may be due to different DWT and DWF conditions in these two studies”.

Lines 236-238: “…It is worth noting that the 13°C + 700 mL/min group had a higher diarrhea index than the 13°C + 300 mL/min group, which may be caused by the greater cold stimulation of the gastrointestinal tract caused by a higher DWF. …” This information is not correct you did not have an interaction.

Response: This sentence has been modified to “The 13°C groups had a higher diarrhea occurrence than the 30°C groups, which may be due to the increased sympathetic nerve excitability and exacerbated bowel motility caused by drinking cold water” .

Conclusion: You should present the conclusion, not repeat the results, just say: Based in our results, piglets weaned in cold environments the water temperature should be set in 30°C and water flow in 300 mL/min.

Response: Thank you so much for the suggestion. The conclusion part has been completely rewritten as follows: “Overall, the present data showed that the DWT should be set at 30°C and DWF at 300 mL/min for weaned piglets in cold environments. This combination of DWT and DWF is beneficial for improving growth performance and feed efficiency, as well as decreasing diarrhea rate in weaned pigs. These beneficial effects may be due to changes in microbial community compositions of the cecum, that led the weaned piglets to have higher utilization efficiency of energy, protein and fat in the feed. Therefore, the combination of drinking water parameters proposed in this study can be used to improve the body weight and health status of weaned piglets in future pig farm management during the cold season”, in line 286-293.

Line 293: You did not evaluate water waste, you cannot conclude it!

Response: The conclusion section has been rewritten and“which is beneficial for reducing water waste”has been removed in line 293.

Reviewer 2 Report

The manuscript is interesting but it needs a deep revision. The main suggestions are: M&M section has to be better described and Results section (and the statistical analysis) has to be reconsidered. Some minor corrections and recommendations are included in the text of the manuscript as comments (in yellow).

Author Response

Thank you very much for the comments and suggestions. We have modified the manuscript accordingly.

Response to Reviewer 2 Comments

Title: “Nutrients” has been changed to “Nutrient”.

Line 52: “Water is necessary for various physiological activities” has been changed to “Water is necessary for several physiological activities”

Line 65: “that” was added after “Recent studies have found”.

Line 68: “In these contexts, improving current knowledge of how different combinations of DWT and DWF...” has been changed to “In these contexts, improving current knowledge of how DWT and DWF...”.

Lines 80-82: The information about gender and crossbreed has been added, “Sixty-four 30-d-old DLY (Duroc × Landrace × Yorkshire) weaned piglets (8.6 ± 0.5 kg, mean ± SD) were randomly divided into four groups with four replicates in each group and four pigs in each replicate, in a balanced sex ratio (2 males and 2 females)”.

Line 80: The variability (SD) has to include a decimal more than the mean.

Response: According to the published papers in Animals, the presentation of SD is consistent with the number of decimal places with the mean, so we think it would be appropriate to stick with the magazine-style.

Line 82: Is it means 2 males and 2 females in each pen?

Response: Yes. The description of gender has been added as “in a balanced sex ratio (2 males and 2 females)”.

Line 86: for piglets of that body weight

Response: “All groups were fed the same diet” was revised as “All groups were fed the same diet for starting pigs”.  

Line 87: The drinker is a key in this manuscript. It has to be well described (type, material, model, company...).

Response: The information about the waterer has been added in lines 87-88, as “ Each pen contained two pig waterers (6.8*2cm. stainless steel, Model WB20XH, Zhenhaidongyi Machinery co. LTD, Ningbo, China).”

Line 94: Remark that diarrhea rate and index were evaluated.

Response: The following sentence has been added in lines 94-97 to note the evaluation of diarrhea rate and index. “The pigs with diarrhea were identified to avoid double counts. Diarrhea rate was calculated by dividing the number of pigs with diarrhea by the total number of pigs in each treatment.The diarrhea index was calculated as [(∑fecal scores for the duration of study)/n]”.

Line 99: No mention before about the sampling of the feeds. In several moments or in just one moment? From one pen or from different pens and after pooled? In the last case, the pool was made per experimental treatment or independent of it?

Response: As all groups were fed the same diet for starting pigs, the feed samples were collected after feed pelleting. The description of feed sampling has been added in lines 99-100 as “Sample of the pelleted feed was taken after its production and stored at -20°C for later digestibility determination.”

Line 101: From each pen? From one animal in concrete? In that case, always the same animal?

Response: Detailed information of fecal sample collection has been added in lines 101-102, “At day 25-28, fresh fecal samples were collected from 4 different locations in each pen floor and mixed in a sterile plastic bag”. They are not from one animal.

Line 105: Stunned? How was the protocol?

Response: Detailed information about anesthetization has been provided in lines 105-106, “The pigs were anesthetized by a combination of azaperone (2 mg/kg body weight) and ketamine (25 mg/kg body weight) administered intramuscularly”.

Line 112: No information is provided about the equipment and analytical methods of the nutrients.

Response: Information about the equipment and methods used to analyze the nutrients has been provided in lines 112-115 as “.Then they were analyzed for crude protein (CP), crude ash, ether extract (EE) according to the following procedures from the Association of Official Analytical Chemists [10]: CP (Method 990.03), ash (Method 942.05) and EE (Method 920.39). The gross energy (GE) were measured in an automatic oxygen bomb calorimeter (PARR 6400; PARR Instruments CO., Moline, IL, US)”.

Line 110: It is an abbreviation and the first time used it has to be described.

Response: The abbreviation AIA stands for “Acid-insoluble ash”, which has been added in line 110.

Line 110: A reference for the protocol of AIA has to be included.

Response:A reference for the AIA protocol has been added as “... Acid-insoluble ash (AIA) was used as an inert marker to determine the apparent digestibility [9]”.

Line 116: “The apparent nutrient digestibility is calculated according to...” has been revised to “The apparent nutrient digestibility was calculated according to ...” to apply correct tense.

Line 118: “A total of ” has been added before “0.3 g of cecum digesta.....”.

Line 124: 95ºC (with no space)Check all through the manuscript! 2 min, Check all through the manuscript!

Response: All spaces between the number and the Celsius symbol have been removed in the text and figures. Spaces have been added between the number and min unit through the whole manuscript.

Line 133: Delete PCA because it is not used thereafter.

Response: The abbreviation PCA has been removed.

Line 145: Authors can inform at the beginning of the Results that no significant interaction was detected in the parameters of Tables 1, 2 and 3 and therefore results will be described by main effects.

Response: Thanks for providing guidance of correct description. The following statement has been added in lines 145-146: “As no significant interaction was detected in the parameters of Tables 1, 2, 3 and 4, results will be described by the main effects.”

Line 149: It is not true. This a trend (p = 0.065) and it has to be described as a trend.

Response: Thanks for pointing out the error in the description. The sentence “ DWT at 30°C increased the final body weight (BW)” has been revised as “DWT at 30°C showed a trend of increasing the final body weight (BW) (p = 0.065) ”.

Line 149:Use the exact p value through the manuscript.

Response: The exact p value has been used in place of “p > 0.05” and “p < 0.05” through the manuscript, except where the descriptions are indicating multiple parameters at the same time.

Line 151: “but had no substantial effect on ADFI ” has been revised to “but no substantial effect was detected on ADFI”.

Line 151: Different letters in the same row have no meaning if P=0.124. Consider no using the post test.....

Response: The superscript letters have been deleted in Table 1 and the description of Table 1 has also been modified as “DWT at 30°C showed a trend of increasing the final body weight (BW) (p = 0.065), and promoted the ADG of weaned piglets considerably (p = 0.043) compared to the 13°C groups. Also, it reduced the F: G ratio (p = 0.045), but no substantial effect was detected on ADFI (p = 0.636)”, in lines 148-151.

Line 152: Avoid the abbreviations in the title of tables.

Response: The abbreviations in the titles of tables have all been changed to full names.

Table1: Always one decimal more than mean.

Response: According to the published papers in Animals, the presentation of SD is consistent with the number of decimal places with the mean, so we think it would be appropriate to stick with the magazine-style.

Line 155: Include here (Table 2). Describe this section by main effects (the performances and the digestibility traits are well described but no in this case). e.g. The diarrhea rate and index with the lower temperature....(because it is irrespective of the flow)

Response: The reference to Table 2 has been moved to the end of this sentence, “Low DWT caused a high incidence of diarrhea” in line154. The description of Table 2 in the paper has been revised as “ The diarrhea rate and index of weaned piglets in the 13°C groups were significantly higher than those in the 30°C groups (p = 0.001 and p = 0.000, respectively). Compared with DWT at 13°C, DWT at 30°C reduced the diarrhea rate of weaned piglets approximately by 50.00% and the diarrhea index by 38.14%. There was no substantial effect of DWF on the diarrhea rate (p = 0.380).” in lines 155-159.

Line 158: “but DWF at 700 mL/min increased the diarrhea index (p < 0.05) when compared with the 300 mL/min groups.” has been removed.

Line 163: No beginning of sentence with an abbreviation. Check through the manuscript.

Response: Thank you for pointing out this error. “DWT at 30°C” has been revised as “The DWT at 30°C”. Other  revisions are in lines 30, 164-165, 207 and 209: “The DWT and DWF were set for ...”, “The DWF had no significant effect on...”, “The DWT change has no ...” and “The DWT at 30°C decreased the relative abundance of ...”, respectively.

Line 164: The location of the table 3 references in this article should be changed.

Response: The reference to Table 3 has been moved to the end of this sentence, “no substantial effect on the digestibility of crude ash (p = 0.266) ”.

Line 165: Delete this sentence.

Response: “There was no substantial interaction between DWF and DWT on apparent nutrient digestibility” in line 165 has been removed.

Line 220: Previous studies? Only one reference is provided.

Response: Thanks for pointing out the grammatical errors. “Previous studies found that pigs...” has been changed to “A previous study found that pigs...”. Only one reference is provided because there are very few published papers on how DWT effects pigs’ growth performance.

Line 237: No reference can justify this finding?

Response: Two references have been cited at the end of the sentence, “...which may be due to the increased sympathetic excitability and exacerbated bowel motility caused by drinking cold water [17, 18]”, in lines 237-238.

Line 284: Describe better the conclusions. It is repeated if the low DWT decreased growth performances and the high DWT promoted growth performances.

Response: Thank you so much for the suggestion. The conclusion part has been rewritten as follows: “Overall, the present data showed that the DWT should be set at 30°C and DWF at 300 mL/min for weaned piglets in cold environments. This combination of DWT and DWF is beneficial for improving growth performance and feed efficiency, as well as decreasing diarrhea rate in weaned pigs. These beneficial effects may be due to changes in microbial community compositions of the cecum, that led the weaned piglets to have higher utilization efficiency of energy, protein and fat in the feed. Therefore, the combination of drinking water parameters proposed in this study can be used to improve the body weight and health status of weaned piglets in future pig farm management during the cold season”, in lines 285-292.

References: Only to remark that there are more papers from the '70s and 80s than from the last 5 years.

Response: Some original older references have been replaced with more recently published papers. Also, some other new references have been cited in the paper due to the content revision.

Reviewer 3 Report

The manuscript presented for the review constitutes and interesting contribution to the science. In fact, the authors have addressed the problem rarely studied, because water in animal nutrition seems to be an obvious and unchangeable element, while its parameters can be expected to influence the rearing or production results.

My suggestions are as follows:

Line 85 - delete “respectively as follows”

I think the authors should include brief  information about the method of diarrhea rate and diarrhea index calculation in “Material and methods”.

Line 96 – what “other clinical symptoms”?

line 106 – “was” instead of “is”

line 152 – “There was no substantial interaction between DWT and DWF on the diarrhea index” – Do the authors mean “There was no substantial effect of DWT and DWF  interaction on the diarrhea index”?

line 161-162 – see previous comment

line 178-179 - see previous comment

Author Response

Thank you very much for the comments and suggestions. We have modified the manuscript accordingly. The attached file is a response to the comments. 

Round 2

Reviewer 1 Report

All my previous suggestion were answered.

Reviewer 2 Report

In my opinion, the manuscript has been notably improved.